# Synergistic Effect of Motivation for the Elderly and Support for Going out

**DOI:** 10.3390/jpm12081257

**Published:** 2022-07-30

**Authors:** Yumi Mashimo, Saki Tsuchihashi, Kenta Tsutsui, Tomoyuki Arai, Yoshitaka Tsuji, Toshiaki Numai, Kazuo Kameda, Kyoko Nishizawa, Mami Kovacs, Shukichi Tanaka, Hiroki Watanabe, Yasushi Naruse, Mitsuyo Ohmura, Noriyuki Ishida, Toshiki Iwasaki, Gaku Hiruma, Naoki Miyazaki, Ryo Takemura, Kengo Nagashima, Yasunori Sato, Yui Ohtsu, Takashi Nakano, Naomi Aida, Isao Iizuka, Hiromi Kato, Yoshiki Kobayashi, Takaaki Senbonmatsu

**Affiliations:** 1Department of Community Medicine, International Medical Center, Saitama Medical University, Saitama 350-1298, Japan; yumimashimo07@yahoo.co.jp; 2Department of Radiology, Saitama Medical University, Saitama 350-0495, Japan; saki_07048@yahoo.co.jp; 3Department of Cardiology, International Medical Center, Saitama Medical University, Saitama 350-1298, Japan; knt22e@gmail.com; 4School of Physical Therapy, Faculty of Health and Medical Care, Saitama Medical University, Saitama 350-0496, Japan; araitm124@gmail.com; 5Division of General Education, Faculty of Health and Medical Care and Department of General Surgery, Saitama Medical University, Saitama 350-1241, Japan; ytsuji@saitama-med.ac.jp; 6Health Promotion Department Long-Term Care Insurance Division, Iruma City Hall, Saitama 358-8511, Japan; ir375000@city.iruma.lg.jp (T.N.); ir113000@city.iruma.lg.jp (K.K.); ir314000@city.iruma.lg.jp (K.N.); mami.kovacs@gmail.com (M.K.); 7Advanced ICT Research Institute, National Institute of Information and Communications Technology, Kobe 651-2492, Japan; tanakas@nict.go.jp; 8Center for Information and Neural Networks, National Institute of Information and Communications Technology, and Osaka University, Kobe 651-2492, Japan; h-watanabe@nict.go.jp (H.W.); y_naruse@nict.go.jp (Y.N.); 9Clinical and Translational Research Center, Keio University School of Medicine, Tokyo 160-8582, Japan; mohmura@keio.jp; 10Biostatistics Unit, Clinical and translational Research Center, Keio University Hospital, Shinjuku, Tokyo 160-8582, Japan; ishida.noriyuki@keio.jp (N.I.); t_iwasaki0521@keio.jp (T.I.); gaku-hiruma@keio.jp (G.H.); nmiyazaki@keio.jp (N.M.); rtakemura@keio.jp (R.T.); nshi@keio.jp (K.N.); 11Department of Preventive Medicine and Public Health, Keio University School of Medicine, Tokyo 160-8582, Japan; yasunori.sato@keio.jp; 12Graduate School of Humanities and Social Sciences, Saitama University, Saitama 338-8570, Japan; ohtsu@mail.saitama-u.ac.jp; 13Kobayashi Hospital, Saitama 358-0014, Japan; kobanakano@gmail.com (T.N.); n.aida@ikkoukai.com (N.A.); yoshikobayashi@ikkoukai.com (Y.K.); 14Business Promotion Department Aisin Co., Ltd., Aichi 448-8650, Japan; isao.iizuka@aisin.co.jp (I.I.); h-kato@drive.aisin.co.jp (H.K.); 15Research Administration Center, Saitama Medical University, Saitama 350-0495, Japan

**Keywords:** motivation, elderly people, subjective well-being, Choisoko

## Abstract

Maintaining a social environment that enables going out freely is important for older people and aids the prevention of frailty syndrome. However, losing a driver’s license can increase the long-term care needs of older people. Therefore, outing support systems are important. However, the utilization rate of these systems is currently relatively low. We conducted a demonstration experiment among older people aged 70 years and over, living in Iruma City, Saitama Japan, by implementing the Choisoko outing support system developed by Aisin Co., Ltd., in conjunction with an approach for improving motivation. Using this system, elderly people were able to go shopping freely whenever they chose, without a driver’s license. Participants in the demonstration experiment exhibited higher Functional Independence Measure scores after the intervention, irrespective of whether or not they used the Choisoko system. The number of uses per person increased over time, and the subjective well-being of Choisoko users improved. However, few male participants engaged with the system. Although improving motivation is important for inducing positive behaviors and enabling the elderly to go out, motivation-improving factors differ between men and women.

## 1. Introduction

The total global population was 7.38 billion in 2015 and is expected to reach 8.5 billion in 2030, 9.7 billion in 2050, and 10.9 billion in 2100 [1]. The percentage of the total population aged 65 years and older increased from 5.1% in 1950 to 8.2% in 2015, and population aging is predicted to progress rapidly in the next half-century [1]. Population aging is expected to progress rapidly not only in advanced countries, but also in developing countries [1].

Comparing the aging rates of advanced countries, Japan had a low rate of aging before the 1980s, was close to the global average in the 1990s, and reached the highest rate of aging of any country in 2005. The rate of aging in Japan is expected to remain at a high level in the future [2].

Although the total population of Japan was 126.17 million in 2019, the mortality rate remains higher than the birth rate, and Japan is entering a long-term depopulation process. By 2029, Japan’s population is predicted to fall below 120 million and continue to decline. By 2053, it is estimated that the population will fall below 100 million to 99.24 million, by 2065, it is predicted to fall to 88.08 million. In this scenario, Japan will become a super-aging society, in which the rate of aging continues to rise as the total population decreases, and the proportion of people aged 65 and over increases [2].

In many suburban areas in Japan, depopulation has resulted in the erosion of regional transportation. In addition, since 2020, the coronavirus pandemic has drastically reduced the opportunities for elderly people to go out, resulting in an increase in sedentary lifestyles that, subsequently, have promoted disuse and increased pressure on social security [3,4]. Public transportation and taxis are abundant in urban areas, and even elderly people who have lost their driver’s license rarely lose the opportunity to go out [5]. However, large cities such as Tokyo and Osaka account for only approximately 20% of the whole population of Japan, and the remaining 80% typically use private cars as transport [6]. In such an environment, losing a driver’s license can make maintaining a social life difficult for older people. For this reason, many local governments operate transportation on demand, such as taxis and community buses for older people whose driver’s licenses have been revoked and take measures to support outing opportunities and to curb disuse by providing long-term care support [7]. However, the participation and retention rates of these schemes are surprisingly low. If the number of users is small, it is difficult for local governments to operate transportation systems. Many elderly people exhibit disuse, causing an increased burden on social security. Despite the existence of outing support systems provided by many local governments, low participation and continuation rates indicate that the target elderly people will rarely use these systems unless their motivation improves. Medical rehabilitation for disuse control typically involves rehabilitation training. Although rehabilitation training is important, if the individual does not enjoy the process, they may lose continuity, and training alone may ultimately be insufficient. We have determined that improving motivation (indicated by sentiments such as “living as a person and having fun on our own” and “living as ourselves”) is a critical issue. Thus, we developed a rehabilitation system to support daily necessities, such as shopping and doctor’s visits, using a transportation service with a strong outing-inducing effect for elderly people [8,9]. Choisoko is an outing support system developed by Aisin Co., Ltd. (Kariya, Aichi 448-8650, Japan) [10]. Many local governments in Japan have implemented the Choisoko system. In this system, participants can make a reservation for transportation by calling the Choisoko Operating Center. This system is similar to a taxi, but if we have another reservation, we pick that person up and drop them off at their destination. The Choisoko system optimizes this process. In addition, the Choisoko system does not impose an economic burden on the user, unlike a taxi. Therefore, we used the Choisoko basic outing support system for elderly people and developed a system to support rehabilitation in daily life. In other words, we developed an outing support system that combines rehabilitation with enabling outings necessary for daily life, to move beyond providing support for older people and provide a tool for preventing frailty. The most important factor in this system is not the rehabilitation itself, but the motivation of elderly people. A social implementation experiment was conducted among elderly people in Iruma City, Saitama Japan.

## 2. Materials and Methods

### 2.1. Subjects

We conducted an observational study of 57 elderly people living in the Miyadera area of Iruma City, Saitama Japan. The inclusion criteria were used to select people aged 70 years or older who had assistance requirements of level 1 or 2, or long-term care requirements of level 1, according to the Long-Term Care Insurance Act in Japan. The long-term care service for elderly under the long-term care insurance system in Japan is divided into seven stages according to the content. Assistance requirements 1 is the mildest state of certification for long-term care and is close to independence in activities of daily living. Assistance requirements 2 is a state that requires a certain amount of social support, although it does not require long-term care. Long-term care 1 is a condition in which elderly people can do most of their personal tasks, such as preparing meals, on their own, but they partially need long-term care. The exclusion criteria for participation were as follows: I, elderly people who could not walk autonomously; II, elderly people with severe illness who had difficulty walking or may be better off; III, elderly people who had difficulty understanding the purpose of the research; and IV, elderly people who did not provide informed consent (Figure 1). The mean age of the subjects was 82.4 years old (82.4 ± 4.58 years old), and the male–female ratio was 15.8% for male subjects and 84.2% for female subjects. Although they did not initially fall under the exclusion criteria, one elderly person was excluded from the study because a serious mental disorder was identified, and we judged that the subject met exclusion criterion III.

### 2.2. Procedures

All procedures in this study were approved by the Ethical Committee of Saitama Medical University (authorization number: Dai 2021-022). Informed consent was obtained from all participants. All measurements were taken at two time points: before Choisoko + rehabilitation effect by going out and 3 months after the start of Choisoko + rehabilitation effect by going out.

### 2.3. Protocol of Transportation and Rehabilitation Effect by Going out

Evaluation of participants’ physical function was conducted using the Functional Independence Measure (FIM), basic check, grip strength, five times rise time, 6 m walking time, and walking speed. FIM is an activities of daily living (ADL) evaluation method developed by Granger et al. in 1983. The FIM can be used to evaluate the burden of long-term care, and is considered to be a reliable and valid method. The FIM is widely used in the field of rehabilitation. The FIM includes 13 items for “exercise ADL,” such as meals and movement, and five items for “cognitive ADL”.

The basic checklist was adapted for the elderly aged 65 and over to check for physical deterioration. This checklist is a tool for preventing the deterioration of older people’s condition by identifying individuals who exhibit a decline in living function at an early stage and implementing approaches for the prevention of long-term care. The checklist consists of 25 questions.

Furthermore, other physical function evaluation methods, including grip strength, five times rise time, 6 m walking time, and walking speed, were measured.

All measurements were taken at two time points: before Choisoko + rehabilitation effect by going out, and 3 months after the start of Choisoko + rehabilitation effect by going out.

### 2.4. Laboratory and Muscle Mass Measurements

Height (cm) and weight (kg) were measured to calculate participants’ body mass index. Body-fat percentage and muscle mass were calculated using a body composition analyzer (InBody770, manufactured by InBody Japan Co., Ltd., Koto-ku, Tokyo 136-0071, Japan). Measurements of iliopsoas muscle volume were performed using computed tomography images taken at the L3–L5 levels by a radiologist. Segmentation was manually performed using free software called ITK-snap, and target areas were extracted.

Using blood samples from all participants, albumin, aspartate aminotransferase (AST)/alanine aminotransferase (ALT), blood urea nitrogen, creatinine for the calculation of estimated glomerular filtration rate, leukocytes, erythrocytes, hemoglobin, hematocrit, mean corpuscular volume (MCV), mean corpuscular hemoglobin (MCH), mean corpuscular hemoglobin concentration (MCHC), and platelets were measured.

All measurements were taken at two time points: before Choisoko + rehabilitation effect by going out, and three months after the start of Choisoko + rehabilitation effect by going out.

A questionnaire was administered to all subjects to record their impressions and opinions regarding Choisoko + rehabilitation effect by going out, and how their quality of life (QOL) changed using the system.

### 2.5. Statistical Analysis

Comparisons between groups were made using Welch’s *t*-test for continuous variables and Fisher’s exact test for categorical variables. The relationship between the number of times the Choisoko system was used and QOL was examined by calculating Spearman’s correlation coefficients. The relationship between the number of times the Choisoko system was used and the change in FIM scores, was also examined by calculating Pearson’s correlation coefficients among participants who used the Choisoko system at least once. Due to the overlap of points, we used a jitter plot to confirm the correlations. We excluded missing data from our analysis. All *p*-values reported in the present study were two-sided and were not adjusted for multiple testing. In this study, *p*-values of less than 0.05 were considered to indicate statistical significance.

All statistical analyses were performed using SAS software version 9.4 (SAS Institute, Inc., Cary, NC, USA).

## 3. Results

### 3.1. Baseline Characteristics

The baseline characteristics of the study participants are presented in Table 1. Of the participants enrolled in this study, 31 used the system and 25 did not use the Choisoko system. The mean ages were 82.9 and 82.1 years old for men and women, respectively, and the median age in both groups was 83.0 years old. There were eight male registrants, six of whom did not use the Choisoko system. In contrast, there were 48 female registrants, 29 of whom used the Choisoko system. Although a significant difference was found in albumin between the participants who used the Choisoko system and those who did not, it is hard to say that there is a statistically clear significant difference considering multiplicity, and both 4.3 g/dL and 4.1 g/dL albumin concentrations are clinically normal values. Therefore, we considered it difficult to correlate this difference with different responses to the presence or absence of the Choisoko system being used as the primary mechanism.

Figure 2 shows the number of Choisoko reservations per month. The number of reservations made by female participants increased steadily every month, whereas the number of reservations made by male participants did not increase at all. The low number of registrations among men may also have affected these results. The average number of uses was 11.9, which exceeded our expectations. The proportion of participants who used the system multiple times may increase over time because people are better able to understand the convenience of the system once they use it.

The Appendix A show a comparison of the amount of change in each measured value between November and December, and between December and February (Appendix A). The Choisoko system was not available until after the first two FIM measurements, to check for differences in FIM scores over time. There was a slight but significant difference in walking time and walking speed between the two groups. This finding suggests that participants may have become accustomed to the testing. However, as expected, no significant differences were found between the first two FIM measurements (Appendix A). In the comparison of the amount of change in each measured value between December and February, basic check, grip strength, five times rise time, 6 m walking time, walking speed, and blood samples did not exhibit significant changes before and after the implementation of the Choisoko system, whether or not the Choisoko system was used, except in the level of body fat. The level of body fat in subjects who used the Choisoko system showed a significant decrease. The amount of activity was not directly measured in this study.

### 3.2. FIM Measurement

Table 2 shows the distribution of total FIM score change related to the Choisoko system in December and February. The difference in the amount of change in FIM scores, with and without the use of the Choisoko system, was 1.39, and the p value was 0.067, indicating a tendency toward a difference. Interestingly, there was an improvement in FIM scores regardless of whether the Choisoko system was used. However, there was no significant difference in the FIM exercise item score change distribution and the FIM cognitive item score change distribution.

Table 3 shows the frequency distribution of the number of times the Choisoko system was used. The results reveal that 61.3% of the subjects used the Choisoko more than 10 times. Most of the users were female, and male participants rarely used the system.

Figure 3A–C show the relationship between the amount of change in the total FIM score and the number of times the Choisoko system was used, the relationship between the amount of change in the FIM exercise score and the number of times the Choisoko system was used, and the relationship between the amount of change in the FIM cognitive score and the number of times the Choisoko system was used. Although the relationship was significant, the effect was relatively small. Figure 4 shows the relationship between the number of times the Choisoko system was used and QOL. The correlation coefficient between these factors was 0.6.

### 3.3. Subjective Well-Being

Figure 5A,B show the results of a questionnaire asking the subjects, who used the Choisoko system, about the relationship between the Choisoko system and their QOL. The results reveal that 83% of Choisoko users said that the Choisoko system clearly increased their opportunities and frequency of going out (Figure 5A). Users also stated that the Choisoko system improved their QOL. The results of the questionnaire reveal that many Choisoko users evaluated the system highly, giving an average score of 4.4 out of 5 points (Figure 5B).

## 4. Discussion

In the current study, we focused on motivation improvement as a critical issue, and incorporated motivational factors in daily life (e.g., shopping, meeting with friends and holding events) into a comprehensive outing support system called the Choisoko system. This system provided rehabilitation effects for older people in daily life, and the results confirm an improvement in FIM scores. Importantly, this intervention relied on the subjects’ freely chosen behavior and did not involve compulsory rehabilitation. The results confirm that FIM scores tended to improve whether or not participants used the Choisoko system. The detailed mechanism of this result is unknown. However, we predict that learning effects can occur in FIM measurement (particularly for cognitive function items). However, even if the current results reflect a learning effect, the improvement may have resulted from participation in the FIM measurement motivating the subjects to obtain higher scores. In addition, although the effect was not statistically significant, subjects who used the Choisoko system showed a greater improvement than subjects who did not. Tijsen et al. identified seven main topics related to challenging rehabilitation environments, which may help to improve rehabilitation outcomes [11]. Seven topics necessary for rehabilitation guidance were selected from *PubMed* and demonstrated to be effective. Although it is also involved in rehabilitation, good communication is an important factor for improving motivation. Monardo et al. reported the evaluation tools available for assessment of patient motivation and satisfaction during technology-assisted rehabilitation (robot rehabilitation, virtual reality rehabilitation, and serious games rehabilitation) [12]. Although Choisoko is an outing support system rather than a technology-assisted rehabilitation method, it may have the effect of increasing the motivation to go out and, consequently, may promote walking exercise. The FIM was originally developed as an index for measuring rehabilitation in subjects with physical impairments, such as stroke patients [13]. Individuals who can walk around their homes and perform most tasks in daily life, such as the subjects in the current study, can typically achieve close to perfect scores in the FIM. Thus, it is possible that a significant difference was not observed because of the ceiling effect, in which the distribution of data is biased to the perfect score. The FIM results in February, among subjects who achieved close-to-perfect FIM scores in December, were almost the same. In contrast, among the subjects who used the Choisoko system, those with low FIM scores in December showed an improvement in FIM scores in February (Appendix A). Nishiwaki et al. reported that motivation is one of the most important factors affecting rehabilitation outcomes [14]. However, this previous study referred to the motivation for performing rehabilitation [15]. Rehabilitation was not compulsory in the current study. Rather than improving the motivation for performing rehabilitation, it may be more important to improve motivation for engaging in daily behavior that has a rehabilitation effect [16]. Thus, even without the professional guidance of a physical therapist, the subject’s motivation to go out or shop may result in a rehabilitation effect.

In this demonstrative experiment, most of the subjects who used the Choisoko system were women. The Choisoko system was used an average of 11.9 times, and approximately 61.1% of subjects used it 10 times or more. One subject used the system 36 times. We expect that once the system is used, the percentage of participants who use it multiple times increases once users understand its convenience and the enjoyment it provides. Supporting this notion, as shown in Figure 1, the number of bookings by female subjects increased every month. In addition, in the latter half of the study, many female subjects booked the Choisoko with friends. For women, friends, shopping, convenience, and events are regarded as potential motivating factors leading them to go out [17]. In contrast, among men, these factors did not lead to any behavioral change. Few men were enrolled in the current study; only two of six men used the Choisoko, and they only used it twice. Identifying motivating factors that induce going out in men may be helpful for promoting significant behavioral changes among the elderly in general. Men may require different motivating factors compared with women [18].

Because the number of times the Choisoko system was used per subject was relatively high, we analyzed the relationship between the number of uses and the other factors measured. Although there was no significant relationship between the amount of change in muscle mass, walking speed, and the amount of change in the number of times the Choisoko was used, the correlation between the amount of change in the number of FIM acquisition points and the amount of change in the number of times the Choisoko was used was 0.35, and the correlation with the amount of change in QOL was 0.6. There was a high correlation between QOL and the number of times the Choisoko was used. Thus, it is possible that environmental changes, which made it possible for participants to freely go wherever they chose without being limited by transportation infrastructure, contributed to the improvement of QOL. In addition, in a questionnaire completed by subjects who used the Choisoko, many users reported that their QOL was significantly improved, and an average score of 4.4 out of 5 points was reported in the questionnaire evaluation method [19,20]. These results indicate that this system has the potential to boost subjective well-being [21,22,23]. However, few male subjects engaged with this system [24,25]. Therefore, as a limitation of the current study, there were insufficient statistical data to analyze the effects of factors such as gender ratio. This issue should be examined in future research.

## 5. Conclusions

While improving motivation is important for inducing behavioral change and enabling the elderly to go out, motivation-improving factors differ between men and women.

## Figures and Tables

**Figure 1 jpm-12-01257-f001:**
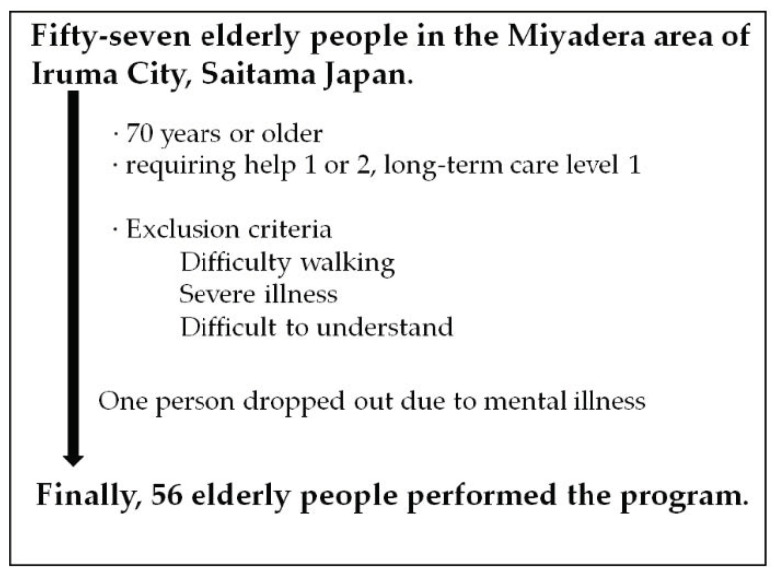
Eligibility of subject person.

**Figure 2 jpm-12-01257-f002:**
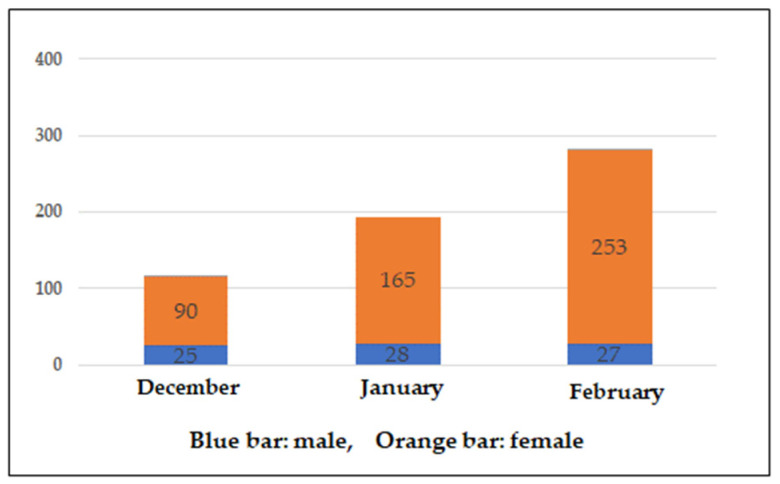
Number of monthly Choisoko reservations. Blue bar indicates male. Orange bar indicates female.

**Figure 3 jpm-12-01257-f003:**
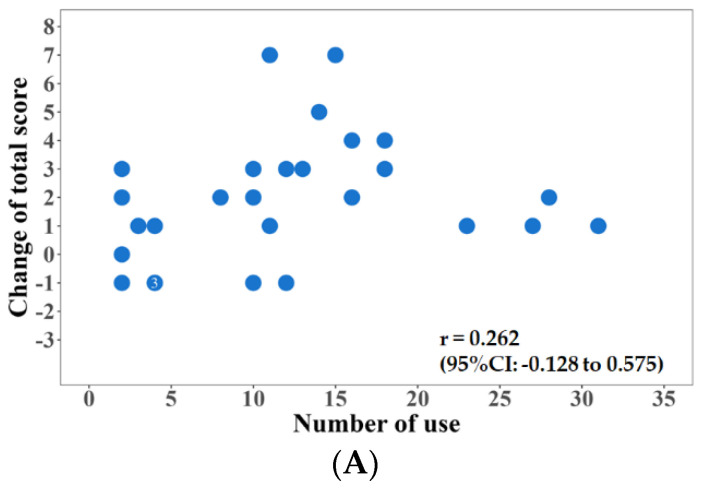
(**A**): relationship between the amount of change in the total FIM score and the number of times the choisoko is used. (**B**): the relationship between the amount of change in the FIM exercise score and the number of times the choisoko is used. (**C**): the relationship between the amount of change in the FIM cognitive score and the number of times the choisoko is used.

**Figure 4 jpm-12-01257-f004:**
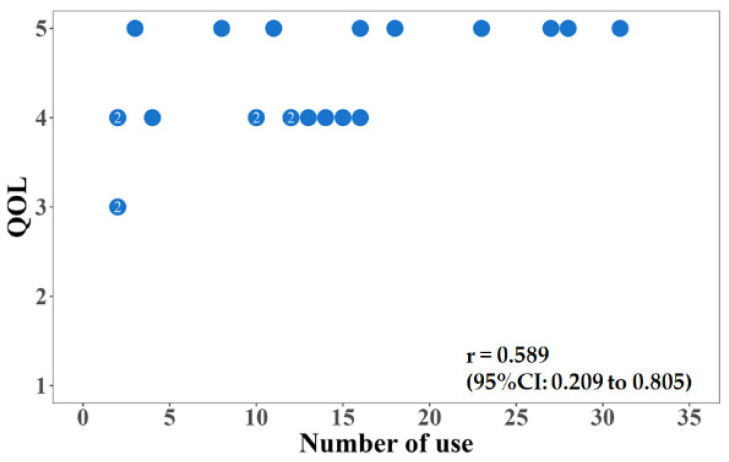
Relationship between number of times the Choisoko using and QOL.

**Figure 5 jpm-12-01257-f005:**
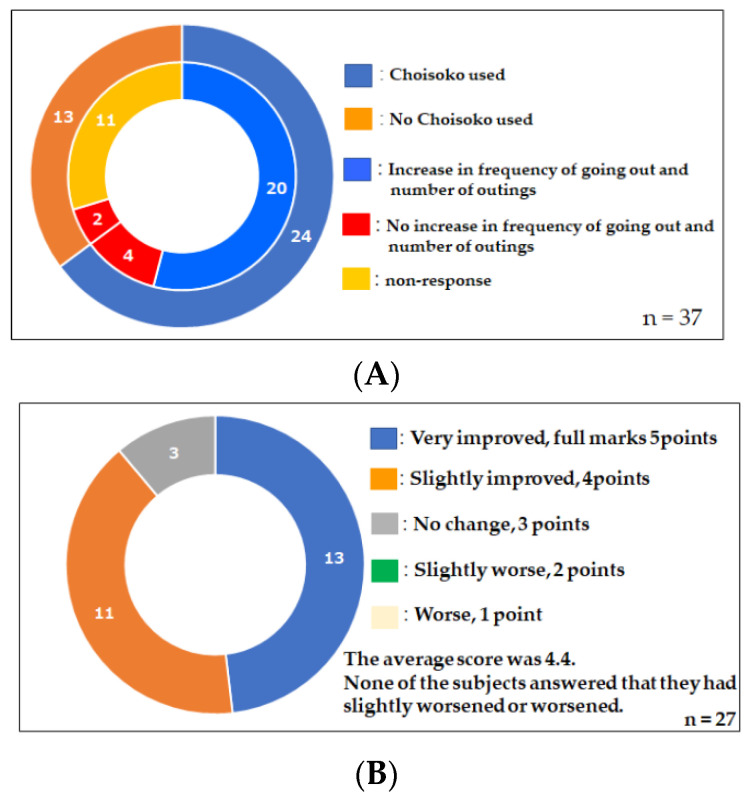
(**A**) Choisoko usage and frequency of going out. The use of Choisoko creates going out. (**B**) Changes in quality of life due to the use of Choisoko.

**Table 1 jpm-12-01257-t001:** Baseline characteristics.

	Choisoko Group (N = 31)		Non Choisoko Group (N = 25)		
	N		N		*p*-Value
Age	31	82.1 ± 4.7	25	82.9 ± 4.6	0.496
Gender (Female)	31	29 (93.5)	25	19 (76.0)	0.069
Height	30	149.5 ± 7.0	24	149.1 ± 7.9	0.847
Weight	30	50.1 ± 8.5	24	49.4 ± 7.5	0.756
Alb	31	4.3 ± 0.3	24	4.1 ± 0.4	0.028
AST	31	26.0 ± 7.5	24	23.8 ± 6.6	0.261
ALT	31	20.0 ± 12.7	24	18.8 ± 7.8	0.667
BUN	31	18.9 ± 5.3	24	17.2 ± 4.1	0.186
CRE	31	0.8 ± 0.2	24	0.8 ± 0.2	0.972
eGFR	31	57.5 ± 12.5	24	59.4 ± 8.9	0.510
WBC	31	5819.4 ± 1446.0	24	6720.8 ± 2203.6	0.091
RBC	31	413.2 ± 41.6	24	408.2 ± 88.3	0.799
Hb	31	12.6 ± 1.3	24	13.0 ± 1.2	0.285
Ht	31	38.1 ± 3.2	24	39.2 ± 3.2	0.225
MCV	31	92.5 ± 5.2	24	92.3 ± 4.4	0.904
MCH	31	30.7 ± 2.3	24	30.6 ± 1.7	0.928
MCHC	31	33.2 ± 1.1	24	33.2 ± 0.8	0.967
Platelet	31	21.7 ± 4.2	24	24.3 ± 5.8	0.075
Grip strength (right)	31	23.3 ± 4.0	25	23.2 ± 4.5	0.876
Grip strength (left)	31	22.1 ± 3.5	25	22.3 ± 4.4	0.888
Grip strength max	31	23.9 ± 3.7	25	24.3 ± 4.4	0.682
Stand up	31	12.0 ± 3.7	24	13.2 ± 7.3	0.486
Walk speed	31	1.0 ± 0.2	24	0.9 ± 0.2	0.147
Walk speed max	31	1.3 ± 0.3	24	1.2 ± 0.3	0.297
Muscle	29	31.7 ± 4.4	24	32.0 ± 4.2	0.800
Quadmus	29	6.0 ± 0.6	25	5.9 ± 0.4	0.469
Volume of iliopsoas muscle	30	4.6 ± 1.8	25	4.4 ± 2.0	0.653

Data are presented as n (%) or mean ± standard deviation. Alb: albumin, AST: aspartate aminotransferase, ALT: alanine aminotransferase, BUN: blood urea nitrogen, CRE: creatinine, eGFR: estimated glomerular filtration rate, WBC: White blood cell, RBC: Red blood cell, Hb: hemoglobin, Ht: hematocrit, MCV: mean corpuscular volume, MCH: mean corpuscular hemoglobin, MCHC: mean corpuscular hemoglobin concentration

**Table 2 jpm-12-01257-t002:** Comparisons of FIM score changes.

	Choisoko Use (N = 31)		Non Choisoko Use (N = 25)			
Change of FIM Score	N	Mean ± SD	N	Mean ± SD	Difference (95%CI)	*p*-Value
Total score	28	1.86 ± 2.24	17	0.47 ± 2.45	1.39 (CI: −0.10 to 2.87)	0.067
Exercise score	28	0.46 ± 1.67	17	−0.41 ± 2.50	0.88 (CI: −0.53 to 2.28)	0.212
Cognitive score	28	1.39 ± 1.52	17	0.88 ± 1.41	0.51 (CI: −0.40 to 1.42)	0.261

SD: Standard deviation, CI: Confidence interval.

**Table 3 jpm-12-01257-t003:** Percentage of reservations per person. n = 31, high percentage of multiple users, because you can understand the convenience once you use it.

Number of Reservations	Number of Target People
1~4	9 (29.0%)	
5~9	3 (9.7%)	
10~19	14 (45.2%)	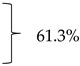
20~29	3 (9.7%)
30 or more	2 (6.4%)

## Data Availability

In this study, the datasets generated and/or analyzed during the current study are available from the corresponding author on reasonable request.

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
