# Peer review of "Synergistic Effect of Motivation for the Elderly and Support for Going out"

_jpm, 2022, doi:10.3390/jpm12081257_

Round 1
Reviewer 1 Report
The authors conducted a clinical interventional study on super-elderlies if emotional support is effective to keep the QOL in Japanese rural dwellings.
I have a concern if the statistical power is calculated to set the sample number.
Author Response
Concern from Reviewer 1.
I have a concern if the statistical power is calculated to set the sample number.
Comment to Reviewer 1.
Thank you for applicable suggestion. This study is an exploratory study, and we had not designed the sample size for the statistical test. We would like to design the sample size in the next demonstration experiment using the parameters drawn from this research.
There was a description error in Table 3, and the values have been corrected.
Reviewer 2 Report
The authors describe a social implementation experiment among 57 elderly people in Iruma City, Saitama Japan, using the Choisoko outing support system. I have a few remarks and suggestions.
Introduction:
- Please avoid terminology promoting the Choisoko system: "comprehensive", "extremely high", etc.
Abstract:
- "Using this system, elderly people were able to go shopping freely whenever they chose, without a driver’s license." -> The focus on shopping might be a reason why there are much less men than women included in the study. I cannot find the information that was given to the participants (informed consent, information letter), but what if its focus was not only on shopping but also on male-oriented social events (e.g. fishing, watching sports, etc.)? This might make a difference as, as stated in the discussion section, " Men may require different motivating factors compared with women". Also, how is the male-female distribution in the general population >= 70 years old in Japan?
Materials & Methods:
- "assistance requirements of level 1 or 2" -> what does this mean? (explain or include reference)
Results:
- "Although a significant difference was found in albumin between male and female participants, we considered this to be an accidental value. " -> Could you explain the reasoning behind this? If it is significant (low p-value?), why would it be accidental?
Discussion:
- line 366: "Data not shown" -> Please make the data available.
Data Availability Statement:
- In this study, it is “Not applicable” here.-> Data should be made available online, so that the results can be reproduced. Also, please make the informed consent and information letter available.
Conflict of interest:
- There are two authors working for Aisin Co. This information should be included here.
Author Response
Concerns from Reviewer 2.
Point 1,
In Introduction, please avoid terminology promoting the Choisoko system: "comprehensive", "extremely high", etc.
Comment to Point 1
We really appreciate your eligible indication. We changed to “Choisoko is an outing support system developed by Aisin Co., Ltd. [10]. Many local governments in Japan have implemented the Choisoko system.”
Point 2
Abstract:
- "Using this system, elderly people were able to go shopping freely whenever they chose, without a driver’s license." -> The focus on shopping might be a reason why there are much less men than women included in the study. I cannot find the information that was given to the participants (informed consent, information letter), but what if its focus was not only on shopping but also on male-oriented social events (e.g. fishing, watching sports, etc.)? This might make a difference as, as stated in the discussion section, " Men may require different motivating factors compared with women". Also, how is the male-female distribution in the general population >= 70 years old in Japan?
Comments to point 2
I really appreciate your suggestion. I agree with your opinion. In this demonstration experiment, we conducted the behaviors necessary for lifestyle-related activities such as shopping to demonstrate the rehabilitation effect of walking for the elderly. I understand that men are not going out at all in this very environment. Currently, the next demonstration experiment is planned, and this result will be reflected. This is described in line 388-391 of the discussion section in the primary version. The ratio of men/women over 70 in Japan is around 0.72.
Point 3,
In Materials & Methods:
- "assistance requirements of level 1 or 2" -> what does this mean? (explain or include reference)
Comment to point 3
Thank you for your great suggestion. This is a system unique to Japan, and it may have been difficult to understand.
The long-term care insurance system in Japan is a system that supports the care of the elderly in society as a whole as the population ages. The long-term care service for elderly under the long-term care insurance system is divided into seven stages according to the content. Assistance requirements 1 is the mildest state of certification for long-term care and is close to independence in activities of daily living. And Assistance requirements 2 is a state that requires a certain amount of social support, although it does not require long-term care. Long-term care 1 is a condition in which elderly can do most of your personal belongings such as meals on his/her own, but elderly partially need long-term care.
Therefore, I modified the following sentences to the manuscript.
We deleted “, which indicated that they could walk around their homes.”
We add “The long-term care service for elderly under the long-term care insurance system in Japan is divided into seven stages according to the content. Assistance requirements 1 is the mildest state of certification for long-term care and is close to independence in activities of daily living. And Assistance requirements 2 is a state that requires a certain amount of social support, although it does not require long-term care. Long-term care 1 is a condition in which elderly can do most of your personal belongings such as meals on his/her own, but elderly partially need long-term care.”
Point 3
Results:
- "Although a significant difference was found in albumin between male and female participants, we considered this to be an accidental value. " -> Could you explain the reasoning behind this? If it is significant (low p-value?), why would it be accidental?
Comment to Point 3
Thank you so much for your eligible suggestion. An error was found in the description in the primary version, and we will correct it. As described in tabel 1, this difference is found in the basic characters of the subjects who used the Choisoko and those who did not. We rewrote it as follows.
Although a significant difference was found in albumin between participants who used the Choisoko system and those who did not, it is hard to say that there is a statistically clear significant difference considering the multiplicity, and both 4.3g/dl and 4.1g/dl of albumin concentration are clinically normal values. Therefore, we considered it difficult to correlate this difference with different responses to presence or absence of Choisoko using as the primary mechanism.
Point 4
Discussion:
- line 366: "Data not shown" -> Please make the data available.
We add the data as the supplement 4.
Comment to Point 4
Thank you so much for your eligible suggestion. We add the result as supplement figure 1.
Point 5
Data Availability Statement:
- In this study, it is “Not applicable” here.-> Data should be made available online, so that the results can be reproduced. Also, please make the informed consent and information letter available.
Comment to Point 5
Thank you for your applicable suggestion. I attached the following sentence. In this study, the datasets generated and/or analyzed during the current study are available from the corresponding author on reasonable request.
Point 6
Conflict of interest:
- There are two authors working for Aisin Co. This information should be included here.
Comment to Point 6
Thank you for your applicable suggestion. We add the following sentence.
Hiromi Kato has no conflict of responsibilities regarding this research.
Isao Iizuka has no conflict of responsibilities regarding this research.
Other authors have no conflict of interest in this research.